# STRATEGIC CLASSIFICATION WITH UNFORESEEABLE OUTCOMES

## ABSTRACT

Machine learning systems are often used to make decisions about individuals, where individuals may best respond and behave strategically to receive favorable outcomes, e.g., they may genuinely *improve* the true labels or *manipulate* observable features directly to game the system without changing labels. Although both behaviors have been studied (often as two separate problems) in the literature, most works assume individuals can (i) perfectly foresee the outcomes of their behaviors when they best respond; (ii) change their features arbitrarily as long as it's affordable, and the costs they need to pay are deterministic functions of feature changes. In this paper, we consider a different setting and focus on *imitative* strategic behaviors with *unforeseeable* outcomes, i.e., individuals manipulate/improve by imitating the features of those with positive labels, but the induced feature changes are unforeseeable. We first propose a novel probabilistic model to capture both behaviors and establish a Stackelberg game between individuals and the decision-maker. Under this model, we examine how the decision-maker's ability to anticipate individual behavior affects its objective function and the individual's best response. We show that the objective difference between the two can be decomposed into three interpretable terms, with each representing the decision-maker's preference for a certain behavior. By exploring the roles of each term, we further illustrate how a decision-maker with adjusted preferences can simultaneously disincentivize manipulation, incentivize improvement, and promote fairness.

## 1 INTRODUCTION

Individuals subject to algorithmic decisions often adapt their behaviors strategically to the decision rule to receive a desirable outcome. As machine learning is increasingly used to make decisions about humans, there has been a growing interest to develop learning methods that explicitly consider the strategic behavior of human agents. A line of research known as *strategic classification* studies this problem, in which individuals can modify their features at costs to receive favorable predictions. Depending on whether such feature changes are to improve the actual labels genuinely (i.e., improvement) or to game the algorithms maliciously (i.e., manipulation), existing works have largely focused on learning classifiers robust against manipulation (Hardt et al., 2016a) or designing incentive mechanisms to encourage improvement (Kleinberg and Raghavan, 2020; Bechavod et al., 2022). A few studies (Miller et al., 2020; Shavit et al., 2020; Horowitz and Rosenfeld, 2023) also consider the presence of both manipulation and improvement, where they exploit the causal structures of features and use *structural causal models* to capture the impacts of feature changes on labels.

To model the interplay between individuals and decision-maker, most existing works adopt (or extend based on) a *Stackelberg game* proposed by Hardt et al. (2016a), i.e., the decision-maker publishes its policy, following which individuals best respond to determine the modified feature. However, these models (implicitly) rely on the following two assumptions that could make them unsuitable for certain applications: (i) individuals can perfectly foresee the outcomes of their behaviors when they best respond; (ii) individuals can change their features arbitrarily at costs, which are modeled as deterministic functions of the feature.

In other words, existing studies assume individuals know their exact feature values before and after strategic behavior. Thus, the cost can be computed precisely based on the feature changes (using functions such as $\ell_p$-norm distance). However, these may not hold in many important applications.

Consider an example of college admission, where the students' exam scores are treated as features in admission decisions. To get admitted, students may increase their scores by either cheating on exams (manipulation) or working hard (improvement). Here (i) individuals do not know the exact values of their original features (unrealized scores) and the modified features (actual score received in an exam) when they best respond, but they have a good idea of what those score distributions would be like from their past experience; (ii) the cost of manipulation/improvement is not a function of feature change (e.g., students may cheat by hiring an imposter to take the exam and the cost of such behavior is more or less fixed). As the original feature was never realized, we cannot compute the feature change precisely and measure the cost based on it. Therefore, the existing models do not fit.

Motivated by the above (more examples are in App. B.2), this paper studies strategic classification with ***unforeseeable outcomes***. We first propose a novel *Stackelberg game* to model the interactions between individuals and the decision-maker. Compared to most existing models (Jagadeesan et al., 2021; Levanon and Rosenfeld, 2022), ours is a *probabilistic framework* that model the outcomes and costs of strategic behavior as random variables. Indeed, this framework is inspired by the models proposed in Zhang et al. (2022); Liu et al. (2020), which only considers either manipulation (Zhang et al., 2022) or improvement (Liu et al., 2020); our model significantly extends their works by considering both behaviors. Specifically, we focus on ***imitative*** strategic behavior where individuals manipulate/improve by imitating the features of those with positive labels, due to the following:

- It is inspired by imitative learning behavior in *social learning*, whereby new behaviors are acquired by copying social models' action behavior. It has been well-supported by literature in psychology and social science (Bandura, 1962; 1978). Recent works (Heidari et al., 2019; Raab and Liu, 2021) in ML also model individuals' behaviors as imitating/replicating the profiles of their social models to study the impacts of fairness interventions.

- Decision-makers can detect easy-to-manipulate features (Bechavod et al., 2021) and discard them when making decisions, so individuals can barely manipulate their features by themselves without changing labels. A better option for them is to hire imposters or steal others' profiles. Such imitation-based manipulative behavior is very common in real world (e.g., cheating, identity theft).

Additionally, our model considers practical scenarios by permitting manipulation to be detected and improvement to be failed at certain probabilities, as evidenced in auditing (Estornell et al., 2021) and social learning (Bandura, 1962). App. A provides more related work and differences with existing models are discussed in App. B.1.

Under this model, we first study the impacts of the decision maker's ability to anticipate individual behavior. Similar to Zhang et al. (2022), we consider two types of decision-makers: non-strategic and strategic. We say a decision-maker (and its policy) is *strategic* if it has the ability to anticipate strategic behavior and accounts for this in determining the decision policies, while a *non-strategic* decision-maker ignores strategic behavior in determining its policies. Importantly, we find that the difference between the decision-maker's learning objectives under two settings can be decomposed into three interpretable terms, with each term representing the decision-maker's preference for certain behavior. By exploring the roles of each term on the decision policy and the resulting individual's best response, we further show that a strategic decision-maker with *adjusted preferences* (i.e., changing the weight of each term in the learning objective) can disincentivize manipulation while incentivizing improvement behavior.

We also consider settings where the strategic individuals come from different social groups and explore the impacts of adjusting preferences on algorithmic fairness. We show that the optimal policy under adjusted preferences may result in fairer outcomes than non-strategic policy and original strategic policy without adjustment. Moreover, such fairness promotion can be attained *simultaneously* with the goal of disincentivizing manipulation. Our contributions are summarized as follows:

1. We propose a probabilistic model to capture both improvement and manipulation; and establish a novel Stackelberg game to model the interplay between individuals and decision-maker. The individual's best response and decision-maker's (non-)strategic policies are characterized (Sec. 2).

2. We show the objective difference between non-strategic and strategic policies can be decomposed into three terms, each representing the decision-maker's preference for certain behavior (Sec. 3).

3. We study how adjusting the decision-maker's preferences can affect the optimal policy and its fairness property, as well as the resulting individual's best response (Sec. 4).

4. We conduct experiments on both synthetic and real data to validate the theoretical findings (Sec. 5).

## 2 PROBLEM FORMULATION

Consider a group of individuals subject to some ML decisions. Each individual has an observable feature $X \in \mathbb{R}$ and a hidden label $Y \in \{0, 1\}$ indicating its qualification state ("0" being unqualified and "1" being qualified).[1] Let $\alpha := \Pr(Y = 1)$ be the population's qualification rate, and $P_{X|Y}(x|1)$, $P_{X|Y}(x|0)$ be the feature distributions of qualified and unqualified individuals, respectively. A decision-maker makes decisions $D \in \{0, 1\}$ ("0" being reject and "1" being accept) about individuals based on a threshold policy with acceptance threshold $\theta \in \mathbb{R}$: $\pi(x) = P_{D|X}(1|x) = \mathbf{1}(x \geq \theta)$. To receive positive decisions, individuals with information of policy $\pi$ may behave strategically by either manipulating their features or improving the actual qualifications.[2] Formally, let $M \in \{0, 1\}$ denote individual's action, with $M = 1$ being manipulation and $M = 0$ being improvement.

**Outcomes of strategic behavior.** Both manipulation and improvement result in the *shifts* of feature distribution. Specifically, for individuals who choose to ***manipulate***, we assume they manipulate by "stealing" the features of those qualified (Zhang et al., 2022), e.g., students cheat on exams by hiring qualified imposters. Moreover, we assume the decision-maker can identify the manipulation behavior with probability $\epsilon \in [0, 1]$ (Estornell et al., 2021). Individuals, once getting caught manipulating, will be rejected directly. For those who decide to ***improve***, they work hard to imitate the features of those qualified (Bandura, 1962; Raab and Liu, 2021; Heidari et al., 2019). With probability $q \in [0, 1]$, they improve the label successfully (overall $\alpha$ increases) and the features conform the distribution $P_{X|Y}(x|1)$; with probability $1 - q$, they slightly improve the features but fail to change the labels, and the improved features conform a new distribution $P^I(x)$. Throughout the paper, we make the following assumption on feature distributions.

**Assumption 2.1.** $P_{X|Y}(x|1), P_{X|Y}(x|0), P^I(x)$ are continuous; both pairs of distributions $\left(P_{X|Y}(x|1), P^I(x)\right)$ and $\left(P^I(x), P_{X|Y}(x|0)\right)$ satisfy the strict monotone likelihood ratio property, i.e. $\frac{P^I(x)}{P_{X|Y}(x|0)}$ and $\frac{P_{X|Y}(x|1)}{P^I(x)}$ are increasing in $x \in \mathbb{R}$.

Assumption 2.1 is relatively mild and has been widely used (Tsirtsis et al., 2019; Zhang et al., 2020a). It can be satisfied by a wide range of distributions (e.g., exponential, Gaussian) and the real data (e.g., FICO data used in Sec. 5). It implies that among qualified (resp. unqualified) individuals and those who improve but fail to change their qualifications, an individual is more likely to be qualified (resp. to be those who successfully improve) as feature value increases. Since $P_{X|Y}(x|1)$ is always the best attainable outcome, only unqualified individuals have incentives to take action (as manipulation and improvement only bring additional cost but no benefit to qualified individuals).

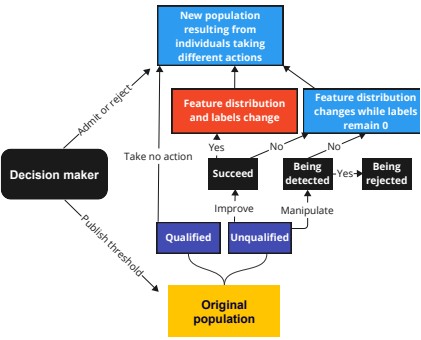

Figure 1: Strategic interaction

### 2.1 INDIVIDUAL'S BEST RESPONSE.

An individual incurs a random cost $C_M \geq 0$ when manipulating the features (Zhang et al., 2022), while incurring a random cost $C_I \geq 0$ when improving the qualifications (Liu et al., 2020). The realizations of these random costs are known to individuals when determining their action $M$; while the decision-maker only knows the cost distributions. Thus, the best response that the decision-maker expects from individuals is the probability of manipulation/improvement. Figure 1 illustrates the strategic interaction between them.

Formally, given a policy $\pi(x) = \mathbf{1}(x \geq \theta)$ with threshold $\theta$, an individual chooses to manipulate only if the expected utility attained under manipulation $U_M(\theta)$ outweighs the utility under improvement $U_I(\theta)$. Suppose an individual benefits $w = 1$ from the acceptance, and 0 from the rejection. Given

---

[1]Similar to prior work (Zhang et al., 2022; Liu et al., 2019), we present our model in one-dimensional feature space. Note that our model and results are applicable to high dimensional space, in which individuals imitate and change all features as a whole based on the joint conditional distribution $P_{X|Y}$ regardless of the dimension of $X$. The costs can be regarded as the sum of an individual's effort to change features in all dimensions.

[2]We assume individuals have budgets to either manipulate or improve. The generalization of considering the actions of "manipulate", "improve", and "do nothing" is discussed in App. B.3.

that each individual only knows his/her label $y \in \{0, 1\}$ and the conditional feature distributions $P_{X|Y}$ but **not** the exact values of the feature $x$, the expected utilities $U_M(\theta)$ and $U_I(\theta)$ can be computed as the expected benefit minus the cost of action, as given below.

$$
\begin{aligned}
U_M(\theta) &= F_{X|Y}(\theta|0) - F_{X|Y}(\theta|1) - \epsilon(1 - F_{X|Y}(\theta|1)) - C_M \\
U_I(\theta) &= F_{X|Y}(\theta|0) - q \cdot F_{X|Y}(\theta|1) - (1-q) \cdot F^I(\theta) - C_I
\end{aligned}
$$

where $F_{X|Y}(x|1)$, $F_{X|Y}(x|0)$, $F^I(x)$ are cumulative density function (CDF) of $P_{X|Y}(x|1)$, $P_{X|Y}(x|0)$, $P^I(x)$, respectively. Given the threshold $\theta$, the decision-maker can anticipate the probability that an unqualified individual chooses to manipulate as $P_M(\theta) = \Pr(U_M(\theta) > U_I(\theta))$, which can further be written as follows (derivations and more explanation details in App. D.1):

$$
P_M(\theta) = \Pr\left((1-q) \cdot \left(F^I(\theta) - F_{X|Y}(\theta|1)\right) - \epsilon\left(1 - F_{X|Y}(\theta|1)\right) \geqslant C_M - C_I\right) \quad (1)
$$

The above formulation captures the imitative strategic behavior with unforeseeable outcomes (e.g., college admission example in Sec. 1): individuals best respond based on feature distributions but not the realizations, and the imitation costs (e.g., hiring an imposter) for individuals from the same group follow the same distribution (Liu et al., 2020), as opposed to being a function of feature changes. equation 1 above can further be written based on CDF of $C_M - C_I$, i.e., the difference between manipulation and improvement costs. We make the following assumption on its PDF.

**Assumption 2.2.** The PDF $P_{C_M - C_I}(x)$ is continuous with $P_{C_M - C_I}(x) > 0$ for $x \in (-\epsilon, 1 - q)$.

Assumption 2.2 is mild only to ensure the manipulation is possible under all thresholds $\theta$. Under the Assumption, we can study the impact of acceptance threshold $\theta$ on manipulation probability $P_M(\theta)$.

**Theorem 2.3** (Manipulation Probability). *Under Assumption 2.2, $P_M(\theta)$ is continuous and satisfies the following: (i) If $q + \epsilon \geqslant 1$, then $P_M(\theta)$ strictly increases. (ii) If $q + \epsilon < 1$, then $P_M(\theta)$ first increases and then decreases, thereby existing a unique maximizer $\theta_{max}$. Moreover, the maximizer $\theta_{max}$ increases in $q$ and $\epsilon$.*

Thm. 2.3 shows that an individual's best response highly depends on the success rate of improvement $q$ and the identification rate of manipulation $\epsilon$. When $q + \epsilon \geqslant 1$ (i.e., improvement can succeed or/and manipulation is detected with high probability), individuals are more likely to manipulate as $\theta$ increases. This is because although individuals are more likely to benefit from improvement than manipulation, as $\theta$ increases to the maximum (i.e., when the decision-maker barely admits anyone), the relative benefit will finally diminish to 0. Thus, more individuals tend to manipulate under larger $\theta$, making $P_M(\theta)$ strictly increasing and reaching the maximum. When $q + \epsilon < 1$, more individuals are incentivized to improve as the threshold gets farther away from $\theta_{max}$. This is because the manipulation in this case incurs a higher benefit than improvement at $\theta_{max}$. As the threshold increases/decreases from $\theta_{max}$ to the minimum/maximum (i.e., the decision-maker either admits almost everyone or no one), the benefit difference between manipulation and improvement decreases to 0 or $-\epsilon$. Thus, $P_M(\theta)$ decreases as $\theta$ increases/decreases from $\theta_{max}$.

## 2.2 DECISION-MAKER'S OPTIMAL POLICY

Suppose the decision-maker receives benefit $u$ (resp. penalty $-u$) when accepting a qualified (resp. unqualified) individual, then the decision-maker aims to find an optimal policy that maximizes its expected utility $\mathbb{E}[R(D, Y)]$, where utility is $R(1, 1) = u, R(1, 0) = -u, R(0, 1) = R(0, 0) = 0$.

As mentioned in Sec. 1, we consider *strategic* and *non-strategic* decision makers. Because the former can anticipate individual's strategic behavior while the latter cannot, their learning objectives $\mathbb{E}[R(D, Y)]$ are different. As a result, their respective optimal policies are also different.

**Non-strategic optimal policy.** Without accounting for strategic behavior, the non-strategic decision-maker's learning objective $\widehat{U}(\pi)$ under policy $\pi$ is given by:

$$
\widehat{U}(\pi) = \int_X \{u\alpha P_{X|Y}(x|1) - u(1-\alpha)P_{X|Y}(x|0)\}\pi(x)\, dx \quad (2)
$$

Under Assumption 2.1, it has been shown in Zhang et al. (2020a) that the optimal non-strategic policy that maximizes $\widehat{U}(\pi)$ is a threshold policy with threshold $\widehat{\theta}^*$ satisfying $\frac{P_{X|Y}(\widehat{\theta}^*|1)}{P_{X|Y}(\widehat{\theta}^*|0)} = \frac{1-\alpha}{\alpha}$.

**Strategic optimal policy.** Given cost and feature distributions, a strategic decision-maker can anticipate an individual's best response (equation 1) and incorporate it in determining its optimal policy. Under a threshold policy $\pi(x) = \mathbf{1}(x \geqslant \theta)$, the objective $U(\pi)$ can be written as a function of $\theta$, i.e.,

$$
\begin{aligned}
U(\theta) \;=\; & u\Big(\alpha + (1-\alpha)(1 - P_M(\theta))q\Big) \cdot \big(1 - F_{X|Y}(\theta|1)\big) \\
& - u(1-\alpha)\Big((1-\epsilon) \cdot P_M(\theta) \cdot \big(1 - F_{X|Y}(\theta|1)\big) + (1 - P_M(\theta)) \cdot (1-q)(1 - F^I(\theta))\Big) \quad (3)
\end{aligned}
$$

The policy that maximizes the above objective function $U(\theta)$ is the strategic optimal policy. We denote the corresponding optimal threshold as $\theta^*$. Compared to non-strategic policy, $U(\theta)$ also depends on $q, \epsilon, P_M(\theta)$ and is rather complicated. Nonetheless, we will show in Sec. 3 that $U(\theta)$ can be justified and decomposed into several interpretable terms.

## 3    DECOMPOSITION OF THE OBJECTIVE DIFFERENCE

In Sec. 2.2, we derived the learning objective functions of both strategic and non-strategic decision-makers (expected utilities $U$ and $\widehat{U}$). Next, we explore how the individual's choice of improvement or manipulation affects decision-maker's utility. Define $\Phi(\theta) = U(\theta) - \widehat{U}(\theta)$ as the *objective difference* between strategic and non-strategic decision-makers, we have:

$$
\Phi(\theta) = u(1-\alpha) \cdot \Big(\phi_1(\theta) - \phi_2(\theta) - \phi_3(\theta)\Big) \tag{4}
$$

where

$$
\begin{aligned}
\phi_1(\theta) \;&=\; \big(1 - P_M(\theta)\big) \cdot q \cdot \big(1 - F_{X|Y}(\theta|0) + 1 - F_{X|Y}(\theta|1)\big) \\
\phi_2(\theta) \;&=\; \big(1 - P_M(\theta)\big) \cdot (1-q) \cdot \big(F_{X|Y}(\theta|0) - F^I(\theta)\big) \\
\phi_3(\theta) \;&=\; P_M(\theta)\big((1-\epsilon)\big(1 - F_{X|Y}(\theta|1)\big) - \big(1 - F_{X|Y}(\theta|0)\big)\big)
\end{aligned}
$$

As shown in  equation 4, the objective difference $\Phi$ can be decomposed into three terms $\phi_1, \phi_2, \phi_3$. It turns out that each term is interpretable and indicates the impact of a certain type of individual behavior on the decision-maker's utility. We discuss these in detail as follows.

1. **Benefit from the successful improvement** $\phi_1$: additional *benefit* the decision-maker gains due to the successful improvement of individuals (as the successful improvement causes label change).

2. **Loss from the failed improvement** $\phi_2$: additional *loss* the decision-maker suffers due to the individuals' failure to improve; this occurs because individuals who fail to improve only experience feature distribution shifts from $P_{X|Y}(x|0)$ to $P^I(x)$ but labels remain.

3. **Loss from the manipulation** $\phi_3$: additional *loss* the decision-maker suffers due to the successful manipulation of individuals; this occurs because individuals who manipulate successfully only change $P_{X|Y}(x|0)$ to $P_{X|Y}(x|1)$ but the labels remain unqualified.

Note that in Zhang et al. (2022), the objective difference $\Phi(\theta)$ has only one term corresponding to the additional loss caused by strategic manipulation. Because our model further considers improvement behavior, the impact of an individual's strategic behavior on the decision-maker's utility gets more complicated. We have illustrated above that in addition to the loss from manipulation $\phi_3$, the improvement behavior also affects decision-maker's utility. Importantly, such an effect can be either positive (if the improvement is successful) or negative (if the improvement fails).

The decomposition of the objective difference $\Phi(\theta)$ highlights the connections between three types of policies: 1) non-strategic policy without considering individual's behavior; 2) strategic policy studied in Zhang et al. (2022) that only considers manipulation, 3) strategic policy studied in this paper that considers both manipulation and improvement. Specifically, by removing $\phi_1, \phi_2, \phi_3$ (resp. $\phi_1, \phi_2$) from the objective function $U(\theta)$, the strategic policy studied in this paper would reduce to the non-strategic policy (resp. strategic policy studied in Zhang et al. (2022)). Based on this observation, we regard $\phi_1, \phi_2, \phi_3$ each as the decision-maker's ***preference*** to a certain type of individual behavior, and define a general strategic decision-maker with adjusted preferences.

### 3.1 Strategic decision-maker with adjusted preferences

We consider general strategic decision-makers who find the optimal decision policy by maximizing $\widehat{U}(\theta) + \Phi(\theta, k_1, k_2, k_3)$ with

$$\Phi(\theta, k_1, k_2, k_3) = k_1 \cdot \phi_1(\theta) - k_2 \cdot \phi_2(\theta) - k_3 \cdot \phi_3(\theta) \tag{5}$$

where $k_1, k_2, k_3 \geqslant 0$ are weight parameters; different combinations of weights correspond to different preferences of the decision-maker. We give some examples below:

1. **Original strategic decision-maker:** the one with $k_1 = k_2 = k_3 = u(1 - \alpha)$ whose learning objective function $U$ follows equation 3; it considers both improvement and manipulation.
2. **Improvement-encouraging decision-maker:** the one with $k_1 > 0$ and $k_2 = k_3 = 0$; it only considers strategic improvement and only values the improvement benefit while ignoring the loss caused by the failure of improvement.
3. **Manipulation-proof decision-maker:** the one with $k_3 > 0$ and $k_1 = k_2 = 0$; it is only concerned with strategic manipulation, and the goal is to prevent manipulation.
4. **Improvement-proof decision-maker:** the one with $k_2 > 0$ and $k_1 = k_3 = 0$; it only considers improvement but the goal is to avoid loss caused by the failed improvement.

The above examples show that a decision-maker, by changing the weights $k_1, k_2, k_3$ could find a policy that encourages certain types of individual behavior (as compared to the original policy $\theta^*$). Although the decision-maker can impact an individual's behavior by adjusting its preferences via $k_1, k_2, k_3$, we emphasize that the **actual utility** it receives from the strategic individuals is always determined by $U(\theta)$ given in equation 3. Indeed, we can regard the framework with adjusted weights (equation 5) as a *regularization* method. We discuss this in more detail in App. B.4.

## 4 Impacts of Adjusting Preferences

Next, we investigate the impacts of adjusting preferences. We aim to understand how a decision-maker by adjusting preferences (i.e., changing $k_1, k_2, k_3$) could affect the optimal policy (Sec. 4.1) and its fairness property (Sec. 4.3), as well as the resulting individual's best response (Sec. 4.2).

### 4.1 Preferences shift the optimal threshold

We will start with the original strategic decision-maker (with $k_1 = k_2 = k_3 = u(1 - \alpha)$) whose objective function follows equation 3, and then investigate how adjusting preferences could affect the decision-maker's optimal policy.

**Complex nature of original strategic decision-maker.** Unlike the non-strategic optimal policy, the analytical solution of strategic optimal policy that maximizes equation 3 is not easy to find. In fact, the utility function $U(\theta)$ of the original strategic decision-maker is highly complex, and the optimal strategic threshold $\theta^*$ may change significantly as $\alpha, F_{X|Y}, F^I, C_M, C_I, \epsilon, q$ vary. In App. C.2, we demonstrate the complexity of $U(\theta)$, which may change drastically as $\alpha, \epsilon, q$ vary. Although we cannot find the strategic optimal threshold precisely, we may still explore the impacts of decision-maker's anticipation of strategic behavior on its policy (by comparing the strategic threshold $\theta^*$ with the non-strategic $\widehat{\theta}^*$), as stated in Thm. 4.1 below.

**Theorem 4.1** (Comparison of strategic and non-strategic policy)**.** *If $\min_\theta P_M(\theta) \leqslant 0.5$, then there exists $\widehat{q} \in (0, 1)$ such that $\forall q \geqslant \widehat{q}$, the strategic optimal $\theta^*$ is always lower than the non-strategic $\widehat{\theta}^*$.*

Thm. 4.1 identifies a condition under which the strategic policy over-accepts individuals compared to the non-strategic one. Specifically, $\min_\theta P_M(\theta) \leqslant 0.5$ ensures that there exist policies under which the majority of individuals prefer improvement over manipulation. Intuitively, under this condition, strategic decision-maker by lowering the threshold (from $\widehat{\theta}^*$) may encourage more individuals to improve. Because $q$ is sufficiently large, more improvement brings more benefit to the decision-maker.

**Optimal threshold under adjusted preferences.** Despite the intricate nature of $U(\theta)$, the optimal strategic threshold may be *shifted* by adjusting the decision-maker's *preferences*, i.e. changing the weights $k_1, k_2, k_3$ assigned to $\phi_1, \phi_2, \phi_3$ in equation 5. Next, we examine how the optimal

threshold can be affected compared to the original strategic threshold by adjusting the decision-maker's preferences. Denote $\theta^*(k_i)$ as the strategic optimal threshold attained by adjusting weight $k_i, i \in \{1, 2, 3\}$ of the original objective function $U(\theta)$. The results are summarized in Table 1. Specifically, the threshold gets lower as $k_1$ increase (Prop. 4.2). Adjusting $k_2$ or $k_3$ may result in the optimal threshold moving toward both directions, but we can identify sufficient conditions when adjusting $k_2$ or $k_3$ pushes the optimal threshold to move toward one direction (Prop. 4.3 and 4.4).

**Proposition 4.2.** *Increasing $k_1$ results in a lower optimal threshold $\theta^*(k_1) < \theta^*$. Moreover, when $k_1$ is sufficiently large, $\theta^*(k_1) < \widehat{\theta}^*$.*

**Proposition 4.3.** *When $\alpha \leqslant 0.5$ (the majority of the population is unqualified), increasing $k_2$ results in a higher optimal threshold $\theta^*(k_2) > \theta^*$. Moreover, when $k_2$ is sufficiently large, $\theta^*(k_2) > \widehat{\theta}^*$.*

**Proposition 4.4.** *For any feature distribution $P_{X|Y}$, there exists an $\bar{\epsilon} \in (0, 1)$ such that whenever $\epsilon \geqslant \bar{\epsilon}$, increasing $k_3$ results in a lower optimal threshold $\theta^*(k_3) < \theta^*$.*

So far we have shown how the optimal threshold can be shifted as the decision maker's preferences change. Next, we explore the impacts of threshold shifts on individuals' behaviors and show how a decision-maker with adjusted preferences can (dis)incentivize manipulation and influence fairness.

Table 1: The impact of adjusted preferences on $\theta^*(k_i)$ compared to original strategic $\theta^*$.

| Adjusted weight | Preference | Threshold shift |
|---|---|---|
| Increase $k_1$ | Encourage improvement | $\theta^*(k_1) < \theta^*$ |
| Increase $k_2$ | Discourage improvement | $\theta^*(k_2) \lessgtr \theta^*$ |
| Increase $k_3$ | Discourage manipulation | $\theta^*(k_3) \lessgtr \theta^*$ |

## 4.2 Preferences as (Dis)incentives for Manipulation

In Thm. 2.3, we explored the impacts of threshold $\theta$ on individuals' best responses $P_M(\theta)$. Combined with our knowledge of the relationship between adjusted preferences and policy (Sec. 4.1), we can further analyze how adjusting preferences affect individuals' responses. Next, we illustrate how a decision-maker may disincentivize manipulation (or equivalently, incentivize improvement) by adjusting its preferences.

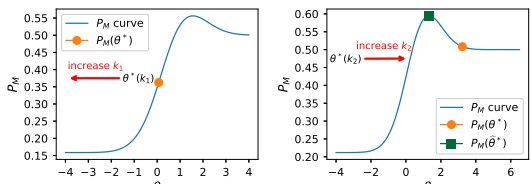

Figure 2: Illustration of scenario 1 (left) and scenario 2 (right) in Thm. 4.5: blue curves are $P_M(\theta)$ and red arrows show how $\theta^*$ changes when adjusting $k_1, k_2$.

**Theorem 4.5** (Preferences serve as (dis)incentives). *Compared to the original strategic policy $\theta^*$, decision-makers by adjusting preferences can disincentivize manipulation (i.e., $P_M(\theta)$ decreases) under certain scenarios. Specifically,*

*1. When **either** of the following is satisfied, and the decision-maker increases $k_1$:*

$$(i). \ q + \epsilon \geqslant 1; \qquad (ii). \ \frac{P_{X|Y}(\theta^*|1)}{P^I(\theta^*)} \leqslant \frac{1-q}{1-q-\epsilon}.$$

*2. When **both** of the following are satisfied, and the decision-maker increases $k_2$:*

$$(i). \ q + \epsilon < 1 \ and \ \alpha < 0.5; \qquad (ii). \ \frac{P_{X|Y}(\theta^*|1)}{P^I(\theta^*)} > \frac{1-q}{1-q-\epsilon} \ and \ P_M(\widehat{\theta}^*) > F_{C_M - C_I}(0).$$

*Moreover, when $k_1$ (for scenario 1) or $k_2$ (for scenario 2) are sufficiently large, adjusting preferences also disincentivize the manipulation compared to the non-strategic policy $\widehat{\theta}^*$.*

Thm. 4.5 identifies conditions under which a decision-maker can disincentivize manipulation directly by adjusting its preferences. The condition $q + \epsilon \lessgtr 1$ determines whether the best response $P_M(\theta)$ is strictly increasing or single-peaked (Thm. 2.3); the condition $\frac{P_{X|Y}(\theta^*|1)}{P^I(\theta^*)} \lessgtr \frac{1-q}{1-q-\epsilon}$ implies that $\theta^*$ is lower/higher than $\theta_{max}$ in Thm. 2.3. In Fig. 2, we illustrate Thm. 4.5 where the left (resp. right) plot corresponds to scenario 1 (resp. scenario 2). Because increasing $k_1$ (resp. $k_2$) results in a lower (resp. higher) threshold than $\theta^*$, the resulting manipulation probability $P_M$ is lower for both scenarios. The detailed experimental setup and more illustrations are in App. C.

## 4.3 Preferences Shape Algorithmic Fairness

The threshold shifts under adjusted preferences further allow us to compare these policies against a certain fairness measure. In this section, we consider strategic individuals from two social groups

$\mathcal{G}_a, \mathcal{G}_b$ distinguished by some protected attribute $S \in \{a, b\}$ (e.g., race, gender). Similar to Zhang et al. (2020a; 2022), we assume the protected attributes are observable and the decision-maker uses *group-dependent* threshold policy $\pi_s(x) = \mathbf{1}(x \geq \theta_s)$ to make decisions about $\mathcal{G}_s, s \in \{a, b\}$. The optimal threshold for each group can be found by maximizing the utility associated with that group: $\max_{\theta_s} \mathbb{E}[R(D, Y) | S = s]$.

**Fairness measure.** We consider a class of group fairness notions that can be represented in the following form (Zhang et al., 2020b; Zhang and Liu, 2021):

$$\mathbb{E}_{X \sim P_a^{\mathcal{C}}}[\pi_a(X)] = \mathbb{E}_{X \sim P_b^{\mathcal{C}}}[\pi_b(X)]$$

where $P_s^{\mathcal{C}}$ is some probability distribution over $X$ associated with fairness metric $\mathcal{C}$. For instance, under equal opportunity (EqOpt) fairness (Hardt et al., 2016b), $P_s^{\text{EqOpt}}(x) = P_{X|YS}(x|1, s)$; under demographic parity (DP) fairness (Barocas et al., 2019), $P_s^{\text{DP}}(x) = P_{X|S}(x|s)$.

For threshold policy with thresholds $(\theta_a, \theta_b)$, we measure the unfairness as $\left| \mathbb{E}_{X \sim P_a^{\mathcal{C}}}[\mathbf{1}(x \geq \theta_a)] - \mathbb{E}_{X \sim P_b^{\mathcal{C}}}[\mathbf{1}(x \geq \theta_b)] \right|$. Define the *advantaged group* as the group with larger $\mathbb{E}_{X \sim P_s^{\mathcal{C}}}[\mathbf{1}(X \geq \widehat{\theta}_s^*)]$ under non-strategic optimal policy $\widehat{\theta}_s^*$, i.e., the group with the larger true positive rate (resp. positive rate) under EqOpt (resp. DP) fairness, and the other group as *disadvantaged group*.

**Mitigate unfairness with adjusted preferences.** Next, we compare the unfairness of different policies and illustrate that decision-makers with adjusted preferences may result in fairer outcomes, as compared to both the original strategic and the non-strategic policy.

**Theorem 4.6** (Promote fairness while disincentivizing manipulation). *Without loss of generality, let $\mathcal{G}_a$ be the advantaged group and $\mathcal{G}_b$ disadvantaged. A strategic decision-maker can always simultaneously disincentivize manipulation and promote fairness in any of the following scenarios:*

1. *When condition 1.(i) **or** 1.(ii) in Thm. 4.5 holds for both groups, and the decision-maker adjusts the preferences by increasing $k_1$ for both groups.*

2. *When condition 2.(i) **and** 2.(ii) in Thm. 4.5 hold for both groups and the decision-maker adjusts the preferences by increasing $k_2$ for both groups.*

3. *When condition 1.(i) or 1.(ii) holds for $\mathcal{G}_a$, condition 2.(i) and 2.(ii) hold for $\mathcal{G}_b$, and the decision-maker adjusts preferences by increasing $k_1$ for $\mathcal{G}_a$ and $k_2$ for $\mathcal{G}_b$.*

Thm. 4.6 identifies *all* scenarios under which a decision-maker can simultaneously promote fairness and disincentivize manipulation by simply adjusting $k_1, k_2$. Otherwise, it is not guaranteed that both objectives can be achieved at the same time, as stated in Corollary 4.7.

**Corollary 4.7.** *If none of the three scenarios in Thm. 4.6 holds, adjusting preferences is not guaranteed to promote fairness and disincentivize manipulation simultaneously.*

The results above assume the decision-maker knows $q, \epsilon$ precisely. In practice, these parameters may need to be estimated empirically. In App. B.5, we further provide an ***estimation procedure*** and present more experimental results when these parameters are noisy.

## 5 EXPERIMENTS

We conduct experiments on both synthetic Gaussian data and FICO score data (Hardt et al., 2016b).

**FICO data (Hardt et al., 2016b).** FICO scores are widely used in the US to predict people's credit worthiness. We use the preprocessed dataset containing the CDF of scores $F_{X|S}(x|s)$, qualification likelihoods $P_{Y|XS}(1|x, s)$, and qualification rates $\alpha_s$ for four racial groups (Caucasian, African American, Hispanic, Asian). All scores are normalized to $[0, 1]$. Similar to Zhang et al. (2022), we use these to estimate the conditional feature distributions $P_{X|YS}(x|y, s)$ using beta distribution $Beta(a_{ys}, b_{ys})$. The results are shown in

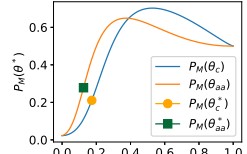

Figure 3: $P_M(\theta)$ of Caucasian and African American.

Fig. 9. We assume the improved feature distribution $P^I(x) \sim Beta\left(\frac{a_{1s} + a_{0s}}{2}, \frac{b_{1s} + b_{0s}}{2}\right)$ and $C_M - C_I \sim \mathcal{N}(0, 0.25)$ for all groups, under which Assumption 2.2 and 2.1 are satisfied (see Fig. 8). We also considered other feature/cost distributions and observed similar results. Note that for

each group $s$, the decision-maker finds its own optimal threshold $\left(\theta_s^* \text{ or } \theta_s^*(k_i) \text{ or } \widehat{\theta}_s^*\right)$ by maximizing the utility associated with that group, i.e., $\max_{\theta_s} \mathbb{E}[R(D, Y)|S = s]$.

We first examine the impact of the decision-maker's anticipation of strategic behavior on policies. In Fig. 23 (App. C.1), the strategic $\theta_s^*$ and non-strategic optimal threshold $\widehat{\theta}_s^*$ are compared for each group under different $q$ and $\epsilon$. The results are consistent with Thm. 4.1, i.e., under certain conditions, $\theta_s^*$ is lower than $\widehat{\theta}_s^*$ when $q$ is sufficiently large.

We also examine the individual best responses. Fig. 3 shows the manipulation probability $P_M(\theta)$ as a function of threshold $\theta$ for Caucasians (blue) and African Americans (orange) when $q = 0.3, \epsilon = 0.5$. For both groups, there exists a unique $\theta_{max}$ that maximizes the manipulation probability. These are consistent with Thm. 2.3. We also indicate the manipulation probabilities under original strategic optimal thresholds $\theta_s^*$; it shows that African American has a higher manipulation probability than Caucasians. Similar results for Asian and Hispanic are shown in Fig. 12.

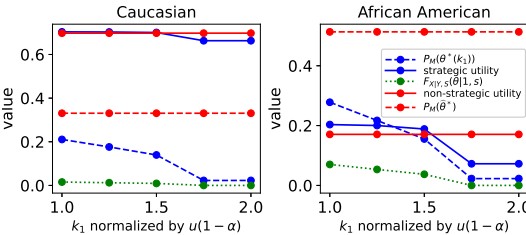

Figure 4: Impact of adjusted preferences (FICO)

Note that the scenario considered in Fig. 3 satisfies the condition *1.(ii)* in Thm. 4.5, because the original strategic $\theta_s^* < \theta_{max}$ for both groups. We further conduct experiments in this setting to evaluate the impacts of adjusted preferences. We first adopt EqOpt as the fairness metric, under which $\mathbb{E}_{X \sim P_s^c}[\mathbf{1}(X \geqslant \widehat{\theta})] = F_{X|YS}(\theta|1, s)$ and the unfairness measure of group $\mathcal{G}_a, \mathcal{G}_b$ can be reduced to $|F_{X|YS}(\theta|1, a) - F_{X|YS}(\theta|1, b)|$. Experiments for other fairness metrics are in App. C.1. The results are shown in Fig. 4, where dashed red and dashed blue curves are manipulation probabilities under non-strategic $\widehat{\theta}^*$ and strategic $\theta^*(k_1)$, respectively. Solid red and solid blue curves are the actual utilities $U(\widehat{\theta}^*)$ and $U(\theta^*(k_1))$ received by the decision-maker. The difference between two dotted green curves measures the unfairness between Caucasians and African Americans. All weights are normalized such that $k_1 = 1$ corresponds to the original strategic policy, and $k_1 > 1$ indicates the policies with adjusted preferences. Results show that when condition *1(ii)* in Thm. 4.5 is satisfied, increasing $k_1$ can simultaneously disincentivize manipulation ($P_M$ decreases with $k_1$) and improve fairness. These validate Thm. 4.5 and 4.6.

Table 2 compares the non-strategic $\widehat{\theta}^*$, original strategic $\theta^*$, and adjusted strategic $\theta^*(k_1)$ when $k_{1,c} = k_{1,aa} = 1.5$. It shows that decision-makers by adjusting preferences can significantly mitigate unfairness and disincentivize manipulation, with only slight decreases in utilities. Results for Asians and Hispanics are in Table 5.

Table 2: Comparison between three types of optimal thresholds (FICO data). For utility and $P_M$, the left value in parenthesis is for Caucasians and the right is for African Americans.

| Threshold | Utility | $P_M$ | Unfairness (EqOpt) |
|---|---|---|---|
| Non-strategic | $(0.698, 0.171)$ | $(0.331, 0.513)$ | $0.136$ |
| Original strategic | $(0.704, 0.203)$ | $(0.211, 0.278)$ | $0.055$ |
| Adjusted strategic | $(0.701, 0.189)$ | $(0.140, 0.155)$ | $0.028$ |

**Gaussian Data.** We also validate our theorems on synthetic data with Gaussian distributed $P_{X|YS}$ in App. C.2. Specifically, we examined the impacts of adjusting preferences on decision policies, individual's best response, and algorithmic fairness. As shown in Fig. 21, 22 and Table 6, 7, 8, these results are consistent with theorems, i.e., adjusting preferences can effectively disincentivize manipulation and improve fairness. Notably, we considered all three scenarios in Thm. 4.5 when condition *1.(i)* or *1.(ii)* or *2* is satisfied. For each scenario, we illustrate the individual's best response $P_M$ in Fig. 21 and show that manipulation can be disincentivized by adjusting preferences, i.e., increasing $k_1$ under condition *1.(i)* or *1.(ii)*, or increasing $k_2$ under condition *2*.

## 6 SOCIETAL IMPACTS & LIMITATIONS

This paper proposes a novel probabilistic framework and formulates a Stackelburg game to tackle imitative strategic behavior with unforeseeable outcomes. The theoretical results depend on some (mild) assumptions and are subject to change when $\epsilon, q, C_M, C_I$ change. Although we provide a practical estimation procedure to estimate the model parameters, it still remains a challenge to estimate model parameters accurately due to the expensive nature of doing controlled experiments. This may bring uncertainties in applying our framework accurately in real applications.

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
