# OpenReview forum: "Strategic Classification with Unforeseeable Outcomes"
_ICLR.cc/2024/Conference — Submitted to ICLR 2024_

### Official Review · Reviewer_7kzc · 2023-10-31

**Soundness:** 2 fair
**Presentation:** 3 good
**Contribution:** 2 fair
**Rating:** 5
**Confidence:** 4

**Summary:**

This paper studies the problem of strategic classification with unforeseeable outcomes, where individuals can improve/manipulate their features by imitating the features of those with possible labels, however, the cost for such manipulation is random, and so is the success of these manipulations. The paper produced an interesting decomposition of the difference between the strategic and non-strategic decision makers into three factors, i.e., benefit from the successful improvement, loss from the failed improvement, and loss from manipulation. Based on this decomposition, the paper proposed a novel decision-maker's utility function as in section 3.1, which captures the different preferences over the aforementioned three factors by different linear weights. The paper shows that these weights can be used as disincentives for manipulation, and promote algorithmic fairness. In addition, the paper produced some explanatory experiments on FICO dataset as well as synthetic Gaussian data.

**Strengths:**

- This model is novel and indeed captures the fact that the outcome of manipulation/improvement is random.
- The presentation of this paper is good, many clean and informative figures are presented, and the notations are well explained.
- The introduction discuss clearly about the rational of assuming the ``imitation behavior".

**Weaknesses:**

- Some of the (main) technical lemmas are adopted from [1].
- This model is a variant of [2], and the subtitle 4.2 ``preferences as (dis)incentives for manipulation" is similar to the title of [2].
- The empirical evaluation needs improvement, i.e., more experiments on different datasets.

Monor:
Wrong Citation:
In the paper: Lydia T. Liu, Sarah Dean, Esther Rolf, Max Simchowitz, and Moritz Hardt. Delayed impact of fair
machine learning. In Proceedings of the Twenty-Eighth International Joint Conference on Artificial
Intelligence, **IJCAI-19**, pages 6196–6200, 2019.

Correct form: Liu, L. T., Dean, S., Rolf, E., Simchowitz, M., & Hardt, M. (2018, July). Delayed impact of fair machine learning. In **International Conference on Machine Learning** (pp. 3150-3158). PMLR.


[1] Zhang, X., Tu, R., Liu, Y., Liu, M., Kjellstrom, H., Zhang, K., & Zhang, C. (2020). How do fair decisions fare in long-term qualification?. Advances in Neural Information Processing Systems, 33, 18457-18469.

[2] Zhang, X., Khalili, M. M., Jin, K., Naghizadeh, P., & Liu, M. (2022, June). Fairness interventions as (dis) incentives for strategic manipulation. In International Conference on Machine Learning (pp. 26239-26264). PMLR.

**Questions:**

See weakness.

---

> ### Author Response · Authors · 2023-11-18
>
> # Response to Reviewer 7kzc
>
> Thanks for your comments. Here are our response point by point:
>
> > Weakness
>
> While [1] also uses a probabilistic framework to model individual behavior, it is limited to manipulative behavior and such manipulation is guaranteed to succeed. In contrast, we propose a more comprehensive probabilistic framework that considers multiple types of individual behaviors (i.e., manipulation and improvement) and each behavior may fail with some probability. Given this framework, we show that the decision-maker's utility can be decomposed into three interpretable terms, and adjusting the weights of these terms can simultaneously promote fairness and disincentive manipulation. The reasoning and technique to approach fairness are entirely different from [1], and the result is novel too.
>
> > Question
>
> Table 2 shows the unfairness of non-strategic threshold is 0.136, the one for original strategic threshold is 0.055 and the one for adjusted is 0.028. So the adjusted threshold has the lowest unfairness (highest fairness).

---

### Official Review · Reviewer_7Bt7 · 2023-10-31

**Soundness:** 2 fair
**Presentation:** 3 good
**Contribution:** 1 poor
**Rating:** 3
**Confidence:** 3

**Summary:**

* This paper discusses a strategic classification with outcome uncertainty and binary action space.
* In the model, each individual has a one-dimensional feature $x\\in\\mathbb{R}$, and a hidden qualification rating $y\\in\\{0,1\\}$. A decision maker decides whether to accept or reject based on a threshold policy $\\pi(x)=1(x \\ge \\theta)$. Unqualified individuals ($y=0$) respond to $\\pi$ by choosing whether to “manipulate” ($M=1$) or by “improve” ($M=0$) their feature to get a better outcome, and qualified individuals do not respond.
* Manipulation is defined as resampling $x$ from the distribution of qualified users $P(x|y=1)$ with probability $1-\\epsilon$, and rejection by the decision maker otherwise. Improvement is defined as sampling from $P(x|y=1)$ with probability $q$, and sampling from $P^I(x)$ otherwise. The random costs associated with manipulation and improvement are $C_M$ and $C_I$, respectively. Each unqualified individual selects $M$ rationally, based on the expected outcome.
* In the theoretical analysis, Theorem 2.3 characterizes the functional form of manipulation probability $P_M(\\theta)$, and Theorem 4.1 provides a provable discrepancy between strategic and non-strategic decision makers. Section 3 decomposes the difference between strategic and non-strategic decision maker utilities into three interpretable components, and propositions 4.2-4.4 demonstrate the effect of changing weights of each component. Finally, Theorem 4.5 identifies conditions under which manipulation can be disincentivized using reweighing of the three components, and Theorem 4.6 identifies conditions under which the decision maker can disincentivize manipulation and promote fairness simultaneously using the same method.
* In the empirical analysis, theoretical findings are validated using parametric distributions (beta distribution fit on FICO data, and synthetic Gaussian data), and the effect of reweighing is explored. Reweighing is shown to significantly decrease the probability of manipulation at the price of a small decrease in accuracy.

**Strengths:**

* Problem is well-motivated.
* Adding outcome uncertainty to the strategic classification framework is an interesting and very natural extension.
* Presentation is clear and easy to follow.

**Weaknesses:**

* There appears to be no discussion about the role of learning from data.
* Some choices in the behavioral model seem implausible, and not properly supported. In particular, it is assumed that qualified individuals that get a negative prediction never make an effort to change it ($y=1, \\pi(x)=0$), and unqualified individuals who get a positive prediction always risk it ($y=0, \\pi(x)=1$) . I didn’t manage to find a discussion of the first concern. For the second concern, Appendix B.3 mentions that the model "can be easily extended" to support a do-nothing action, but the actual extension and its consequences are not described.
* Theoretical results are obtained under strong assumptions, namely one-dimensional space and monotone likelihood ratio. It is not clear how they extend to more complex settings.

**Questions:**

* Which model parameters are assumed to be known to the decision maker, and how are unknown parameters estimated from data?
* What information is needed by the individual in order to make a rational decision?
* How would the conclusions change in cases where rejected qualified users are given an option to improve/manipulate, and accepted unqualified agents have an option to maintain their features?
* What is the optimal trade-off between strategic utility $U(\\theta)$ and manipulation probability $P_M(\\theta)$? Is it attained by the preference adjustment method?

---

> ### Author Response · Authors · 2023-11-18
>
> # Response to Reviewer 7Bt7
>
> Thanks for your comments. Here are our response point by point:
>
> > Weakness 1: learning from data
>
> Please see App.B.5 where we provide a complete estimation procedure. With only the knowledge of conditional distribution of qualified individuals $P_{X|Y}(x|1)$ and the population's qualification rate $\alpha$, we introduce a complete procedure to estimate $P_{X|Y}(x|0), q, P^I, \epsilon, P_{C_M-C_I}(x)$ sequentially in App.B.5.
>
>
>
> > Weakness 2: Some choices in the behavioral model seem implausible
>
> We believe there is a misunderstanding of this point. An agent with $y =0$ will not know whether she will get a positive outcome ($\pi(x)$) when she needs to make a decision to manipulate/improve. This is the key modeling of our paper. Thus, she can only apply probabilistic knowledge to determine whether to improve/manipulate, and all unqualified agents will make decisions according to Eqn. (1).
>
>
>
> > Weakness 3: Theoretical results are obtained under strong assumptions, namely one-dimensional space and monotone likelihood ratio. It is not clear how they extend to more complex settings
>
>
>
> Firstly, monotone likelihood ratio is a common assumption and has been used under most strategic classification settings ([Hardt et al., 2016; Raab & Liu, 2021]). Secondly, our model focuses on settings where individuals manipulate or improve to mimic the profiles of qualified ones; it is not limited to one-dimensional feature setting. Specifically, in high-dimensional space, individuals need to change features in all dimensions, and manipulation/improvement cost can be regarded as the sum of effort/investment an individual makes to change all features. For example, an individual who choose to manipulate needs to manipulate multiple features to mimic a qualified individual’s features; manipulation of each feature can induce some cost (which may or may not be correlated) and the overall effect is captured by the sum of all component costs, which is the total manipulation cost in our model. We will add the above explanation and present the results for general setting in the revision.
>
>
>
> > Question 1: Which model parameters are assumed to be known to the decision maker, and how are unknown parameters estimated from data?
>
> Please see response to weakness 1.
>
> > Question 2: What information is needed by the individual in order to make a rational decision?
>
> The individual knows $P_{X|Y}, \epsilon, q, C_M, C_I$ to make a rational decision. Note that a large line of literature assumed individuals have perfect knowledge, so the assumption should not be too strong. Moreover, in App.B.5, we also provide complementary experiments under noisy response.
>
> > How would the conclusions change in cases where rejected qualified users are given an option to improve/manipulate, and accepted unqualified agents have an option to maintain their features?
>
> Thanks for the insightful question. Since we already clarify that the agents will not know their decision outcomes when they have to make a decision, so this question is only valid when we consider a sequential setting to understand the long-term dynamics of agents’ qualifications. This can be a meaningful direction for future research.
>
> > Question 3: What is the optimal trade-off between strategic utility and manipulation probability?
>
> To find the optimal trade-off, the decision-maker can first define a metric to measure the trade-off, then run a grid search on $k_1, k_2, k_3$ to determine the best parameter configuration.

---

> > ### Comment · Reviewer_7Bt7 · 2023-11-21
> >
> > Thank you for your response! I have no further questions.

---

### Official Review · Reviewer_q7Ka · 2023-11-01

**Soundness:** 3 good
**Presentation:** 3 good
**Contribution:** 3 good
**Rating:** 5
**Confidence:** 3

**Summary:**

This paper studies a novel strategic classification setting, which combines both manipulations and improvements on the agents' part, as well as a strategic decision maker who is free to adjust some regularization parameters to control for either type of behavior. Unlike in many existing setups, here each individual in the dataset can either manipulate the features without improving its qualification, or elect to improve the qualification --- both at distinct costs. Moreover, once again distinct from most standard settings, choosing to manipulate or improve does not lead to a deterministic change in e.g. outcome once the corresponding cost is paid; instead, there is randomness in the system, such that an improver will with some probability $q$ fail to improve the qualification, and similarly a manipulator will with some probability $\epsilon$ be detected and banned from participation.

It is shown in this setting that: (1) depending on whether $q + \epsilon \geq 1$ or not, the probability of any population member being manipulative is either monotonically increasing in the decision threshold $\theta$, or first increases and then decreases. (2) The strategic decision maker's utility function at each threshold $\theta$ is a sum of three terms: one that reflects benefit from improvement; the second one that reflects loss from failed improvement; and the third one that reflects loss coming from manipulation. (3) By letting the decision maker toggle these three terms by multiplying them by factors $k_1, k_2, k_3$, they define and study what is essentially a "regularized" decision maker, who by toggling one of $k_1, k_2, k_3$ can, up to certain assumption, (a) lower or increase decision threshold $\theta$; and (b) disincentivizing manipulation.

There is also an application to fairness, whereby the decision maker can mitigate unfairness (demographic parity, equal opportunity and some more general definition) in a two disjoint groups setting while at the same time disincentivizing manipulative behavior. Finally, two experiments are conducted, one on FICO data and another one on synthetic data, showing that the theoretical conclusions proved in the paper are true in practice as well.

**Strengths:**

The main strength of the paper is fourfold:
(1) its combined handling of manipulations and improvements (compared to prior works such as Zhang et al. that handle these separately);
(2) letting a decision maker control both effects given a threshold policy, together with theoretical results to that extent;
(3) the randomized, rather than deterministic, nature of anticipated outcomes that come with improvement or manipulation;
(4) the ability for the decision maker to explicitly control other statistics regarding deploying policies on the data.

**Weaknesses:**

There are no particular weaknesses that I could observe; but some parts of the framework and technical aspects deserve further elaboration, as in the questions below.
(1) Generally, more meat on the bone in terms of practical applications of the proposed framework --- with corresponding experiments --- would be good to have.
(2) As well as giving more concrete rates of improvement in e.g. manipulation probabilities etc. that the decision maker could obtain by increasing/decreasing the k's.

**Questions:**

I have some questions on the theorem statements/results/model formulation in the main part of the paper:

1. I am having a hard time parsing the intended setting of p_I --- the distribution that unsuccessful improvers fall into. For instance, in one of the experiments, it is set to a distribution with averaged successful/unsuccessful parameters. However, it is not a priori clear what this distribution should represent in various non-synthetic potential applications, and if this modeling aspect is even justified that often.

2. The statements of the main results are obtained essentially by taking derivatives and evaluating their signs; I would like to see a more concrete discussion, in the main part, of the actual rates at which the various quantities of interest improve/decay. This might help at least partially address the following practical question:

3. What are the normative takeaways? The framework is interpreted by the authors, among other things, as a way to add regularization from the decision maker's perspective to be able to achieve desired effects on improvement and manipulation probabilities, group fairness, etc. However, with an abundance of ways to set k1, k2, k3 to get different effects, it is not clear what course of action the decision maker should take in a given setting. As in the above question, examining the sensitivity of the resulting downstream effects to the choice of the k's appears to be one way to begin to understand this; but also I would like to see some high-level discussion on this for potential applications.

---

> ### Author Response · Authors · 2023-11-18
>
> # Response to Reviewer q7Ka
>
> Thanks for your comments. Here are our response point by point:
>
> > The modeling of $P_I$
>
> In all results of the main paper, $P_I$ only needs to satisfy assumption 2.1, which is reasonable without restricting the specific distributions. Though we assume $P_I$ as a prior knowledge, but in practice, researchers may need to gather data through controlled experiments to get $P_I$. Just as [Miller et al., 2020] pointed out, strategic classification in fact needs causal modeling.
>
> > Normative takeaways of adjusting preferences
>
> The normative takeaways of how to adjust preferences mainly rely on Thm. 4.6. The decision-maker can first distinguish whether either of scenarios (I)-(III) is satisfied, thereby choosing the way to adjust the preferences accordingly. This will ensure the decision-maker can always simultaneously disincentivize manipulation and promote fairness.

---

> > ### Comment · Reviewer_q7Ka · 2023-11-22
> > **Response to Authors**
> >
> > This is to acknowledge that I've read both the authors' response to my question, as well as all other discussion in the thread. I've reassessed my point of view somewhat, and now believe that it is suboptimal to have a combination of the following three features of the paper: (1) the qualitative nature of the theoretical results (meaning, as pointed out in the above review, that they mostly say something like 'the derivative is positive here subject to conditions XYZ, so when XYZ holds, increasing k1/k2/k3 results in improved W', but by how much to increase the ki's is relegated to e.g. grid search), (2) the stylized nature of the proposed model (as pointed out in several reviews), and (3) very limited empirical evaluation.
> >
> > The authors' responses to everyone have in my opinion not added too much extra clarity on the modeling, or new compelling empirical applications of the framework. There are many ways this can be done in the future --- for one starting point, the way the framework is motivated in the intro, it would be nice to actually design/conduct experiments where impersonating other people takes place, and argue/justify whether the presented framework is indeed an interesting way to capture such a setting.
> >
> > In sum, given the other reviews and my own reflection, I've reconsidered and will slightly lower my score from 6 to 5; I still maintain that the paper has some strengths as discussed above, but would in hindsight have appreciated more effort in the direction of convincing the readers of the theoretical/practical usefulness of the framework as it is modeled here.

---

### Official Review · Reviewer_5Dsj · 2023-11-06

**Soundness:** 3 good
**Presentation:** 3 good
**Contribution:** 2 fair
**Rating:** 3
**Confidence:** 3

**Summary:**

This paper considers a principal-agent problem in which the principal—an ML-based classifier—makes a decision about an agent in the style of strategic classification. The agent may alter the features observed by principal through either manipulation or improvement, in possibly a non-deterministic manner, and the principal may or may not be aware of this strategic behavior. With this model, the authors study the effect of the principal being aware of strategic behavior, the impact of the principal's preferences, and experimental simulations of their model.

**Strengths:**

- This paper ventures their modeling into real-world aspects overlooked in typical models of strategic classification, such as uncertainty about the impact of the agent’s actions and imitative learning of other agents' behavior.
- Solid technical contributions in modeling and analysis, with clear writing throughout.

**Weaknesses:**

- From my reading of the paper, it is not clear to me what the key qualitative takeaways from this model/analysis are. It would be helpful if the authors could elaborate a bit more on this aspect of their work.
- In a similar vein, I find that the presentation of the model is overly focused on the technical aspects rather than the modeling motivations of the assumptions. It would help the paper if the model were simplified to focus on what is truly needed to get to the kernel of the insight.
- As principal-agent problems are a cross-disciplinary topic, the paper would also benefit from discussion of the study of such principal-agent problems beyond the recent work in computer science. For instance, manipulation of observed features has been studied as far back as Spence (*Quarterly Journal of Economics*, 1973) and more recently by Frankle and Kartik (*Journal of Political Economy*, 2019) and Ball (working paper, 2022).

**Questions:**

- Are there simpler special cases that still provide interesting insights?
- What are the core qualitative insights revealed by this framework?

---

> ### Author Response · Authors · 2023-11-18
>
> # Response to Reviewer 5Dsj
>
>
>
> Thanks for your comments. Here are our response point by point:
>
>
>
> > Key takeaways
>
>
>
> - The first contribution of our work is that we first propose a fundamentally different model of strategic classification to deal with the unforeseeable nature of the outcomes of strategic actions and can capture both manipulation and improvement behaviors.
>
>
>
> - We also reveal how the decision-maker can adjust its preferences to achieve both fairness and disincentivize manipulation, thereby shedding light on how to make socially responsible decisions (Thm 4.6).
>
>
>
> > Simpler case
>
>
>
> Our model can indeed handle simpler cases where only strategic manipulation or improvement is available for agents. When $q = 0$, the model only permits manipulation with some detection probability; When $\epsilon = 1$, the model only permits improvement. However,  we aim to provide a comprehensive probabilistic framework to give insight to the decision-maker even when the agents display diverse behaviors.

---

### Meta-Review · Area_Chair_86gT · 2023-12-11

**Metareview:**

The paper studies a new model for strategic classification that directly captures the uncertainty strategic agents may have on how they could influence their outcomes or observable features. The reviewers identified several weaknesses with the current submission. Most notably, the paper could better articulate its key qualitative takeaways and core insights. Relatedly, the authors could consider providing stronger motivations for the model’s assumptions.

**Justification For Why Not Higher Score:**

There is a unanimous consensus for not accepting the paper.

**Justification For Why Not Lower Score:**

N/A

---

### Decision · Program_Chairs · 2024-01-16

Reject